# State-of-the-Art Review: Technical and Imaging Considerations in Hybrid Transcatheter and Minimally Invasive Left Ventricular Reconstruction for Ischemic Heart Failure

**DOI:** 10.3390/jcm11164831

**Published:** 2022-08-18

**Authors:** Romy Roosmarijn Maria Jacqueline Josepha Hegeman, Martin John Swaans, Jan-Peter van Kuijk, Patrick Klein

**Affiliations:** 1Department of Cardiothoracic Surgery, Sint Antonius Hospital Nieuwegein, 3435 CM Nieuwegein, The Netherlands; 2Department of Cardiology, Sint Antonius Hospital Nieuwegein, 3435 CM Nieuwegein, The Netherlands

**Keywords:** hybrid left ventricular reconstruction, ischemic heart failure, ischemic cardiomyopathy, left ventricular remodeling, minimally invasive cardiac surgery

## Abstract

Negative left ventricular (LV) remodeling consequent to acute myocardial infarction (AMI) is characterized by an increase in LV volumes in the presence of a depressed LVEF. In order to restore the shape, size, and function of the LV, operative treatment options to achieve volume reduction and shape reconstruction should be considered. In the past decade, conventional surgical LV reconstruction through a full median sternotomy has evolved towards a hybrid transcatheter and less invasive LV reconstruction. In order to perform a safe and effective hybrid LV reconstruction, thorough knowledge of the technical considerations and adequate use of multimodality imaging both pre- and intraoperatively are fundamental. In addition, a comprehensive understanding of the individual procedural steps from both a cardiological and surgical point of view is required.

## 1. Introduction

Ischemic heart disease (IHD) is the single most common cause of death worldwide [1] and can lead to ischemic heart failure (IHF) if it remains undiagnosed or untreated [2]. Early reperfusion after acute myocardial infarction (AMI) by either pharmacological means (thrombolysis) or primary percutaneous coronary intervention (PCI) can limit the size of infarction and preserve left ventricular (LV) function [2]. However, not all patients with AMI improve or maintain cardiac function [3] due to negative LV remodeling [4]. Negative LV remodeling consequent to acute myocardial infarction (AMI) is characterized by an increase in LV volumes in the presence of a depressed LVEF due to increased wall tension, disrupted myocardial fiber orientation, and the presence of myocardial scar tissue [5]. As end-systolic and end-diastolic volumes (ESV and EDV) increase after the index event, LV systolic performance declines [6]. This mechanical adaptation to declining systolic function initially facilitates preservation of stroke volume but eventually turns into HF when the rightward end of the pressure-volume relationship, as described by Frank and Starling, is reached [6]. In order to restore the shape and size of the LV, operative treatment options to achieve volume reduction should be considered. In the past decade, conventional surgical LV reconstruction through a full median sternotomy has evolved towards a hybrid transcatheter and less invasive LV reconstruction, also known as the less invasive ventricular enhancement (LIVE) procedure. With this review, we sought to provide an extensive overview of preprocedural imaging, anatomical and technical considerations, and patient outcome of the LIVE procedure.

## 2. The History of LV Reconstruction Surgery

The first surgical ventricular aneurysm (VA) repair was performed by Claude Beck in 1944 by attaching a fascia lata graft to the external surface of the anterior VA [7,8]. In 1956, Bailey reported the repair of four VAs without the use of extracorporeal circulation by first placing a large toothed clamp across the base of the VA and then excising and repairing the VA [7]. Cooley was the first surgeon who used a pump oxygenator in 1958 to excise a VA and repair the ventricle with linear closure [7,9]. It took several decades until the long-term experience of 523 patients who underwent a ventricular aneurysmectomy was published by Coltharp in 1994, describing four different techniques: linear, septal, purse-string, and patch technique [7]. This study revealed that patients with good contractility of the non-aneurysmal LV had better long-term survival than patients with an impaired non-aneurysmal LV [7]. Hereafter, a trend developed towards a more physiological and anatomical reconstruction of the residual ventricular cavity. Consequently, Stoney revealed a repair technique in 1973 in which the free lateral ventricular wall of the aneurysm is brought down and sewn to the scar along the septum, enabling exclusion of a- or dyskinetic septal scar [7,10]. A similar physiologic elimination of septal scar was reported by both Dor and Cooley in 1989, with the incorporation of a prosthetic Dacron patch in the closure of the defect post aneurysmectomy instead [11,12]. The technical caveats as addressed by Coltharp [7], namely injuring coronary artery supply and compromise of diastolic volume due to extensive ventriculotomy and scar excision, initiated further development of techniques leading up to the Dor procedure in which an endoventricular patch was used, thus avoiding a large resection [13]. With the Dor procedure, the LV is reshaped with a stitch that encircles the transitional zone between contractile and noncontractile myocardium, and a small patch is placed to reestablish ventricular wall continuity at the level of the purse-string suture [14]. However, this ‘functional amputation’ of the LV with the exclusion of the entire a- or dyskinetic scar led to increased chamber sphericity and a suboptimal short axis/long axis ratio, possibly influencing the development of late mitral regurgitation [15]. Finally, the use of a preshaped elliptical balloon (Chase Medical, Dallas, TX, USA) has been added to the procedure to help reconfigure the ventricle and prevent a compromised diastolic volume [16].

## 3. Transition from Conventional Surgical LV Reconstruction to an Alternative Hybrid Approach

Surgical ventricular reconstruction (SVR) has been demonstrated to be an effective therapy in selected patients with ischemic HF in expert centers [17]. However, conventional SVR is a highly invasive open-heart surgical procedure that requires a full median sternotomy with the use of extracorporeal circulation (ECC) and cardioplegic myocardial arrest [5]. Therefore, a transition has taken place towards less invasive surgical off-pump LV reconstruction. Early experience with the first-generation BioVentrix Epicardial Revivent TC System was first reported by Wechsler et al. in 2013 [4]. The Epicardial Revivent^TM^ System enabled LV reconstruction on the beating heart [17] without the need for ventriculotomy [6] by apposition of the scarred lateral LV wall to the septal scar with paired anchors placed through epicardial transmural catheters [18]. As this procedure was still performed through a full median sternotomy, the technique further evolved towards a less invasive and more hybrid approach with the use of transcatheter techniques. In 2016, the second-generation Revivent TransCatheter^TM^ (TC^TM^) System received CE marking certification. The so-called Less Invasive Ventricular Enhancement (LIVE) procedure enables deployment of the titanium anchor pairs through a hybrid approach by combining access to the beating heart via a left-lateral mini-thoracotomy with transcatheter access via the right internal jugular vein (RIJV) [17].

## 4. LIVE Therapy Stitching Technique: What to Stitch to Achieve Optimal LV Reconstruction?

The LIVE procedure aims to reconstruct the LV by plication of fibrous scar in order to reduce the enlarged LV volume, decrease wall stress and increase the EF [4]. The LIVE procedure is based on the micro-anchoring technology of the Revivent TC^TM^ Ventricular Enhancement System (BioVentrix Inc., San Ramon, CA, USA), consisting of multiple paired anchors that are connected by a poly-ether-ether-ketone (PEEK) tether [4]. The internal and external micro-anchors are brought together over the PEEK tether, forming a longitudinal line of apposition between the LV free wall and the anterior septum of the right ventricle (RV) [4]. Dependent on the distribution of myocardial scar tissue, LIVE therapy with Revivent TC includes different optional approaches: Hybrid RV-LV with or without external LV-LV and/or external RV-LV (Figure 1).

### 4.1. Septal Scar (Limited or Extended)

Septal localization of myocardial scar requires a hybrid RV-LV approach. The internal anchor (from the RIJV) with its’ corresponding external anchor is the first pair to be implanted. This can be repeated several times.In case more septal scar is present basal to this pair/these pairs, an external RV-LV anchor pair can be placed. This external RV-LV stitch (“Antonius stitch”) is a modification of the hybrid approach, in which the anchor pair is deployed fully externally instead of a combination of internally (with transcatheter techniques through the RIJV) and externally.If scar tissue is present in the apex, additional external LV-LV anchor pairs should be placed and can be combined, if necessary, with a LV purse-string suture.

### 4.2. External Approach for Septal Scar in High-Risk Patients

When the endovascular approach is considered high risk and/or not feasible, an external-only approach can be applied [19]. If both septal and apical myocardial scar distributions are present, external RV-LV anchors (“Antonius stitch”) and external LV-LV anchors should be placed. A LV purse-string suture can then also be added to the procedure.

### 4.3. Isolated Apical Aneurysm

When there is a clear apical aneurysm with anterolateral scar in the absence of septal scar distribution, an LV-purse string suture should be placed together with external LV-LV anchor pairs.

## 5. The Role of Pre- and Post-Operative Imaging Modalities in the LIVE Procedure

Adequate imaging is of great importance in order to evaluate patient-specific indications and contra-indications [5] and contribute to procedural planning. Detailed and comprehensive pre-operative imaging of patients with ischemic heart failure should include transthoracic echocardiography (TTE), cardiac magnetic resonance (CMR) imaging with late gadolinium enhancement (LGE), and/or four-dimensional computed tomography [20].

### 5.1. Transthoracic Echocardiography

TTE is utilized to systematically assess LV dilatation, diminished LVEF, LVESVI, and LVEDVI, regional wall akinesis and/or dyskinesis and valvular incompetence (especially functional mitral valve regurgitation (FMR)) [21]. More specifically, the biplane Simpson method should be used for the determination of the LVEF, which should be derived from measurements in the AP4CH and AP2CH views [22]. In addition to standard volumetric measurements, the LV outflow tract velocity time integral (LVOT VTI) can be used to quantify cardiac systolic function and cardiac output (CO) [23]. The LVOT VTI should be derived from AP3CH or AP5CH views with pulsed wave Doppler, measured approximately 0.5 cm proximal from the aortic valve. The stroke volume (LVOT area (Figure 2) x LVOT VTI) and CO can then be calculated with the use of the LVOT VTI but are only reliable in the absence of aortic regurgitation [22].

As the SV, CO, and CI are influenced by the LV pre- and afterload, it is useful to additionally determine the LV contractility. The LV contractility is measured in the isovolumetric contraction phase and is, therefore, independent of loading conditions. LV contractility represents the ratio of pressure change in the ventricular cavity during the isovolumetric contraction period (dP/dt) and can be determined with the use of the CW-signal of the MI [22].

Furthermore, the LV sphericity index (LVSI) should be calculated as LV short-to-long axis ratio, derived from a four-chamber view. The apical conicity index (apical-to-short axis length ratio in a four-chamber view) can also be determined [24,25]. In extensive global dilation of the LV after a large MI, the LVSI regularly increases. However, this is not the case for isolated apical LV aneurysm formation, in which the LVSI remains stable, but the ACI increases [24,25].

### 5.2. Cardiac Magnetic Resonance (CMR) Imaging with Late Gadolinium Enhancement (LGE)

When procedural eligibility has been confirmed by initial TTE evaluation, LGE CMR imaging should be performed as it gives a higher resolution image, precisely determines the location and transmurality of myocardial scar, and assesses myocardial viability [5,20,21]. Gadolinium-based contrast accumulates within areas of increased extracellular space (such as fibrosis), thereby enhancing scarred areas [26]. The patterns of LGE help to differentiate between ischemic and non-ischemic myocardial injury since ischemic myocardial injury causes typical subendocardial or transmural LGE [26]. The cardiac LV segmentation model described by the American Heart Association (AHA) facilitates the description of scarred myocardium based on the distribution of LGE by dividing the heart into 17 segments [26,27]. These segments are consistent with vascular supply from the three major coronary arteries [26,27]. A transmural scar is characterized by a hyperenhancement extending ≥51% of the LV wall thickness in ≥1 of the LV segments [26]. In addition to scar quantification, LGE CMR provides accurate measures of LV volumes and ejection fraction [5,21,28,29]. The LVSI can also be calculated using the ratio of the LVEDV to the volume of a sphere with the long-axis LVEDD [28]. Moreover, the presence of intracardiac thrombus (Figure 3) can be ruled out with 88% sensitivity [5,20]. Of note, admission of gadolinium-based contrast should be avoided in patients with advanced renal dysfunction (glomerular filtration rate < 30 mL/min/1.73 m^2^) [20].

### 5.3. Four-Dimensional Cardiac Computed Tomography

A four-dimensional cardiac computed tomography (4D CT) scan with a triphasic injection of contrast can be a suitable alternative to LGE CMR for patients with intracardiac devices that are incompatible with CMR, such as Internal Cardioverter Defibrillators and pacemakers [5,21]. Although 4D CT does not have the same scar determination capabilities due to its moderate diagnostic accuracy in the detection of MI (sensitivity 52–78%, specificity 88–100%) [20], it can determine regional wall thickness and motion abnormalities [5]. Thinned, akinetic, or dyskinetic regions are considered non-viable and are therefore suitable for exclusion with the LIVE procedure [5]. A clear benefit of cardiac CT over CMR is the ability to assess LV wall calcification, which cannot be depicted from CMR [5]. Moreover, graft location and patency can be evaluated in patients who underwent coronary artery bypass grafting [5]. The presence of intraventricular thrombus can be detected by CT but is of inferior sensitivity and specificity compared to LGE CMR [5].

### 5.4. Three-Dimensional Reconstructive Imaging

CT-derived three-dimensional reconstructions (Figure 4) can visualize the ventricular aneurysm and can complement procedural planning of anchor placement [5,17].

## 6. A Guide through the LIVE Technique (Procedural Steps)

The LIVE procedure is performed by a collaboration of a cardiothoracic surgeon (CTS) and interventional cardiologist (IC) [4]. Support of a skilled imaging cardiologist is of the essence, as intraoperative transesophageal echocardiography (TEE) is required to provide procedural assistance, monitor LV, and valve function, and evaluate myocardial anchor implantation [21].

### 6.1. Patient Set-Up for the LIVE Procedure

The procedure is performed under general anesthesia with a single-lumen endotracheal tube. External defibrillator pads and meticulous control of activated clotting time (ACT) add to patient safety. In order to create easy access for the mini-thoracotomy, the left shoulder and elbow should be slightly flexed dorsally.

### 6.2. Location of Incision

First, a 6 cm left antero-lateral mini-thoracotomy is performed by the CTS. The optimal intercostal space (ICS) is identified with the use of TTE. The left pleural cavity is usually accessed through the 4th or 5th ICS [4].

### 6.3. Insertion of the Endovascular Snare System in the RV

A 14 Fr Introducer is inserted by the IC in the RIJV and advanced into the superior vena cava or right atrium. A Swan–Ganz catheter is advanced through the 14 Fr sheath into the pulmonary artery (PA), through which a 0.025 Jagwire is inserted. The 14 Fr sheath is also advanced into the RV towards the interventricular septum. The Swan–Ganz catheter is exchanged for a 7 or 8 Fr multipurpose catheter (MP) and the 0.025″ wire is left in the PA. The multi-looped EnSnare^TM^ endovascular snare system is advanced through the MP catheter and deployed in the RV apex. Meanwhile, an RV gram is performed to confirm proper positioning [21].

### 6.4. Insertion of the Needle through the RV Epicardium towards the Septum

A 10 cm long 18-gauge straight needle is inserted on the scarred epicardium (1–2 cm under the LAD) towards the selected septal site as directed by the EnSnare in the RV apex. The septal puncture location has been identified pre-operatively with pre-operative MRC or CT. Needle position in the RV is validated with contrast-injection within the needle, after which a 0.032″ wire is advanced through the needle into the RV [21]. After a 6 Fr dilator is placed over the 0.032″ wire towards the RV, the 0.032″ wire is withdrawn, and a 5 Fr Amplatz Right Mod Catheter (AR) is advanced to the RV, directing a 0.018″ guidewire into the RV towards the multi-looped snare [4].

### 6.5. Meeting in the Middle

The IC then removes the 0.025″ Jagwire, and the snare is pulled back to grab the 0.018″ wire. The AR is advanced by the CTS. Then, the snare, 0.018″ guidewire, and the MP are withdrawn through the 14Fr introducer by the IC. The CTS simultaneously advances the AR through the LV into the 14 Fr introducer all the way towards the RIJV, while the IC keeps retracting the MP through the 14 Fr introducer [4].

### 6.6. Introduction of the Internal Anchor

Consequently, the 0.018″ wire is exchanged for a 0.014″ wire, over which the internal anchor assembly is advanced by the IC from the RIJV. The 0.014″ wire should be kept under tension by the CTS and IC. The internal anchor assembly is advanced into the 14 Fr introducer, and the CTS pulls in synchrony. Once the internal anchor reaches the RV septum, the 14 Fr introducer is slightly withdrawn to reveal the anchor [4].

### 6.7. Placement of the External Anchor

The external anchor is placed over the tether and advanced to the epicardial surface. Tension should be maintained on the anchor tether so that the internal anchor remains positioned on the RV septum [4].

### 6.8. Imaging-Guided Release of the Anchor

Hemodynamics, LV configuration, and tricuspid valvular function are assessed by TEE and fluoroscopy. If appropriate, the anchor is released against the right side of the septum [21].

### 6.9. Hybrid Placement of Additional Anchor Pairs

The steps are repeated for each LV-RV anchor pair [21], with a typical consecution from apical to the basal septum.

### 6.10. Additional External RV-LV Anchor Pair(s)

When an additional basal anteroseptal scar is present, external RV-LV anchor pairs—also called “Antonius stitches”—can be added. This acronym is used since this technique was developed in the St. Antonius Hospital (Nieuwegein, The Netherlands). The concept of the reconstructive technique is the same as with the hybrid approach: Anterolateral scar is brought into contact with (antero)septal scar, but the difference with these “stitches” is that the anchor-pair is deployed fully externally. This prevents any intracardiac manipulation such as snare deployment or snaring of guidewires and hence reduces the risk for ventricular arrhythmias and hemodynamic disturbances. A Prolene-1 suture with a large TP-1 needle is placed from the anterolateral LV wall, passed through the anteroseptum below de LAD, and then out through the free RV wall. Hereafter, a 4Fr dilator is placed over this suture, creating a protected tube through the heart, through which a 0.014” guidewire is then placed. Over this guidewire, a (flex-)anchor on a tether is placed over a large felt pledget and passed reversely from the free RV wall, through the anterior septum, and out the anterolateral wall. The tether is then cut, and a mating external anchor (also over large felt pledget) deployed under pressure guidance. The felt pledgets ensure hemostasis and prevent rupture through a thin and potentially friable free RV wall.

### 6.11. Additional External LV-LV Anchor Pair(s)

When the final anchor is LV-LV (i.e., at the LV apex with no further space in the RV apex), the LV needle should be driven across the cardiac apex, preferably from right to left, with visibility gained from manipulation of previous anchor pairs [21]. After the needle tip is retrieved, the hinged anchor should be placed on the right side of the apex. The tether can be cut at or near the needle to allow the placement of an external anchor for hemostasis.

All anchor positions are checked fluoroscopically for alignment.

### 6.12. Closure

The pericardium is left completely open. Routine closure in layers is performed with a single small caliber left pleural drain (21Fr). A vessel closure device (such as Proglide or Angioseal) is used for the closure of the arterial puncture site in the groin.

## 7. Insights from Previous Literature

### 7.1. Finding the Right Patient: Suggested Indications for Hybrid LVR

The in- and exclusion criteria that were applied by the previous case series [17,19,29,30,31] are summarized in Table 1. Patients were considered eligible for hybrid transcatheter and minimally invasive LVR when they presented with symptomatic HF corresponding to NYHA class II or higher as a consequence of cardiac dysfunction due to previous anterior MI [17,19,29,30,31]. The LV was dilated with LVESVI ≥ 60 mL/m^2^ (reported in three studies [17,19,32]) with a reduced ejection fraction below 45% due to akinesia or dyskinesia in the anteroseptal, apical, or apicolateral region [17,19,29,30,31]. In order to provide sufficient support to the anchor pairs, transmural scarring (typically ≥ 50%) is necessary [17,19,29,30,31]. Patients maintained guideline-directed medical therapy for ≥90 days and were optimized on stable target dosages [17,19,31,32]. This means that the AMI or index event occurred at least three months before screening, which is important because sufficient scar maturation or fibrotic tensile strength is necessary to provide support and prevent wall lesion due to tearing of the anchors through the wall [5]. Logically, the viability of myocardium in regions remote from the area of intended scar exclusion is of the essence but was not specifically reported in all studies.

### 7.2. Finding the Right Patient: Suggested Contraindications for Hybrid LVR

The main contraindications reported by previous studies [17,19,29,30,31] were the presence of an intracardiac thrombus, myocardial infarction within 90 days before the procedure, systolic pulmonary arterial pressure > 60 mmHg on echocardiography and contraindication to open-heart surgery in case of conversion after the occurrence of a complication. Important valvular lesions that necessitated repair or replacement were also a contraindication [17,19,33]. Other exclusion criteria that were reported by one or more studies were: Significant diastolic dysfunction, calcification of the ventricular wall in the area of intended scar exclusion, cardiac resynchronization therapy (CRT) device placement ≤60 days prior to screening, functioning pacemaker leads in the antero-apical right ventricle that would interfere with anchor placement, previous sternotomy, previous left-sided thoracotomy and contraindication for anticoagulation therapy [17,19].

### 7.3. Included Studies and Baseline Characteristics of Described Cohorts

Up until today, the outcome of hybrid LVR has been described in a pooled amount of 100 patients (Table 2), including three monocenter and two multi-center studies [17,19,29,30,31]. Pooled data revealed that the majority of patients were male (81%) with a mean age of 61 ± 11 years across groups (Table 3). Patients had symptomatic HF corresponding to NYHA class 2.6 ± 0.5. Average pre-operative LVEF was 31 ± 8%, LVESVI was 76 ± 27 mL/m^2^, and LVEDVI was 108 ± 33 mL/m^2^. Concomitant PCI was performed in 5 out of 35 patients in the multi-center study performed by Klein et al. [17].

### 7.4. Procedural Data and Clinical Outcome of Hybrid LVR

Procedural data of all published series are tabulated in Table 4. A total of 100 patients were included, of whom 99 eventually underwent a hybrid LVR [17,19,29,30,31]. Mean operating time was 218 ± 74 min. Conversion to full median sternotomy was necessary in three patients (3%) across groups due to RV perforation (*n* = 2) and acute mitral regurgitation caused by chordal impairment (*n* = 1). Three patients (3%) required re-intervention. One patient underwent post-operative revision because of RV restriction, during which the external anchor of an RV-LV anchor pair was removed, with full recovery of the patient as a result [30]. Two other patients were subjected to conventional reoperation after three days and six weeks post-operatively because of increasing tricuspid valve regurgitation [29]. Procedure-related mortality was 6% across groups (*n* = 6). Reasons of death were: Stroke (*n* = 2), shock with multiple organ failure (*n* = 2), pulmonary vascular complication (*n* = 1), and esophageal rupture with sepsis (*n* = 1). Four patients died during reported follow-up in all studies combined, adding up to a total all-cause mortality rate until the latest reported follow-up across groups of 10% (*n* = 10). Reasons of death during follow-up were: Lung-carcinoma (*n* = 1), alcohol abuse (*n* = 1), bacteremia after infected diabetic foot (*n* = 1), and sudden death (*n* = 1).

### 7.5. Echocardiographic Outcome of Hybrid LVR

A significant reduction of both LVESVI and LVEDVI was found in all studies at all reported moments of follow-up up to five years post-operatively [17,19,29,30,31].

Naar et al. reported the longest follow-up with a decrease in LVESVI from 75 ± 25 mL/m^2^ to 50 ± 20 mL/m^2^ at two years and 56 ± 16 mL/m^2^ five years post-operatively (*p* < 0.001; *p* = 0.047) [29]. Klein et al. reported a LVESVI decrease from 75 ± 32 to 50 ± 12 (*p* < 0.001) at a one-year follow-up [33]. Wang and colleagues showed a reduction in LVESVI from 85 ± 26 to 66 ± 24 mL/m^2^ after nine months post-operatively [31]. Shorter follow-up revealed a decrease from 93 ± 11 to 52 ± 15 (*p* < 0.001) and from 53 ± 8 to 30 ± 11 (*p* < 0.001) at discharge and directly post-operatively [19,30].

A significant reduction of LVEDVI was similarly described from 110 ± 39 to 78 ± 19 at a 12-month follow-up (*p* = 0.015) [33], from 108 ± 33 to 91 ± 32 at 9 months [31], from 137 ± 20 to 78 ± 10 (*p* = 0.001) at discharge [19] and from 75 ± 23 to 45 ± 6 directly post-operatively [30].

A significant improvement in LVEF was found in four out of five studies [17,19,29,30,31]. In the study of Naar et al., improvement in LVEF was significant only at a two-year follow-up [29]. Early post-operatively, a clear increase of LVEF from 23 ± 8% to 35 ± 7% (*p* = 0.001) and 28 ± 8% to 40 ± 10% (*p* < 0.001) was reported by Loforte et al. and Klein et al. [19,30]. Wang and colleagues reported a significant improvement of LVEF from 36 ± 9% to 46 ± 10% (*p* < 0.001) at nine months post-operatively [31]. In the relatively larger cohort of Klein et al., a non-significant increase in LVEF from 30 ± 8 to 36 ± 6 was described at a 12-month follow-up.

In the patient population described by Naar et al., progression of tricuspid regurgitation (TR) was observed compared to baseline grade 0.6 ± 0.6 (scale 0–4): 1.68 ± 0.8 after 6 months (*p* < 0.001), 1.18 ± 0.8 after 2 years (*p* = 0.008), and 1.65 ± 1.0 after 5 years (*p* = 0.003) [29]. Two of the patients of this cohort were re-operated due to deteriorating TR [29]. In the relatively smaller cohorts described by Klein et al. and Loforte et al., a mild increase to moderate TR was described in two patients and one patient had a decrease from moderate to mild TR [19,30].

### 7.6. Functional Outcome of Hybrid LVR

Long-term follow-up of operated patients was performed in one study by Naar et al. [29]. All studies reported an improvement in NYHA class, with a significant reduction described in four out of five studies [17,19,29,30,31]. Naar and colleagues reported a significant improvement from NYHA class 2.3 ± 0.5 to 1.6 ± 0.7 (*p* = 0.01) at a five-year follow-up [29]. Klein et al. described a NYHA class reduction of 2.6 ± 0.5 to 1.7 ± 0.6 at 12 months (*p* < 0.001) in the largest cohort until today (*n* = 35) [17]. Wang demonstrated a reduction of NYHA class from 2.7 ± 0.6 to 1.7 ± 0.7 nine months post-operatively [31]. Loforte et al. reported a reduction from 3.4 ± 0.6 to 1.4 ± 0.9 in 7 patients (*p* = 0.001) [19].

In the three largest cohorts, a six-minute walk test (6-MWT) was performed during follow-up and revealed a significant improvement over time [17,31,32]. Pre- and post-operative total walking distance as reported by Naar et al., Klein et al. and Wang et al. improved from 381 ± 103 m, 365 ± 90 m and 369 ± 40 m to 432 ± 77, 450 ± 75 and 462 ± 61 m respectively at two years; 12-month and 9-month follow-up (*p* = 0.02; *p* = 0.002; *p* < 0.001) [17,31,32].

Quality of life based on the Minnesota Living with Heart Failure Questionnaire (MLHFQ) was described in two studies [29,33], with a significantly improved score of 39 vs. 26 points (*p* < 0.001) after 12 months in one of two studies [33].

## 8. Conclusions

Up until today, five case series (ranging from 7 to 35 patients per cohort) have described the outcome of hybrid LVR with the use of the Revivent TC System. This technique has been shown to be a safe and effective minimally invasive treatment option to reconstruct the negatively remodeled LV after AMI. In addition to achieving a significant improvement in both LV volume indices and ejection fraction up to five years post-operatively in the majority of patients, a significant functional improvement was obtained in terms of six-minute walk test and NYHA class.

## Figures and Tables

**Figure 1 jcm-11-04831-f001:**
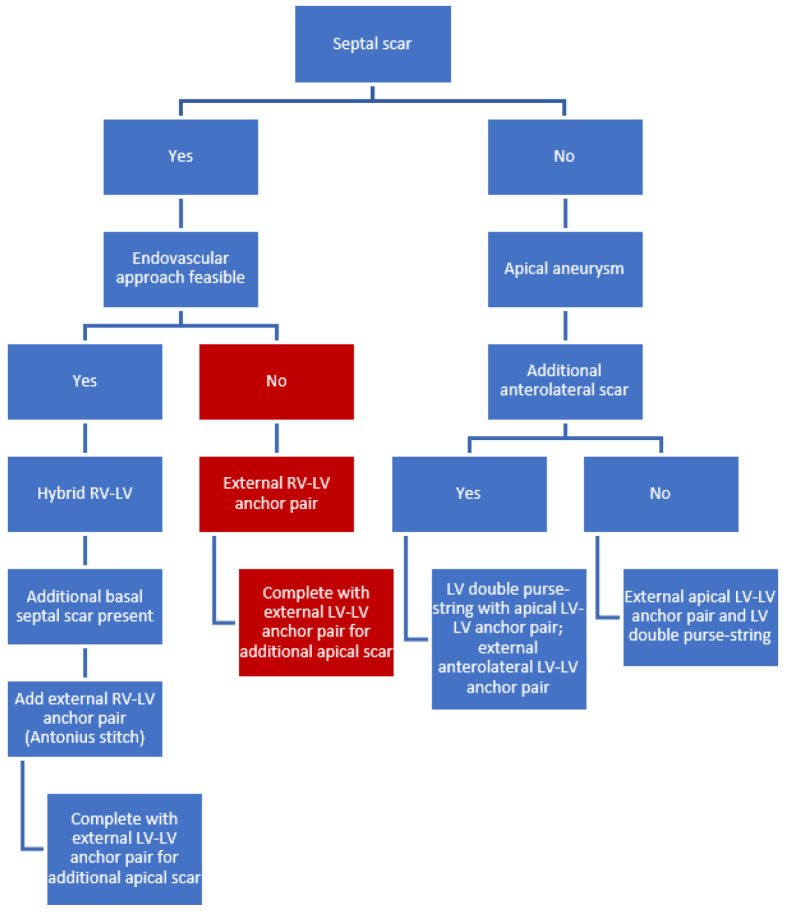
Flowchart of technical considerations based on scar distributions. Colored in red is the bail-out procedure in case the indicated hybrid endovascular approach is considered non-feasible because of increased procedural risk. Abbreviations: LV, left ventricle; RV, right ventricle.

**Figure 2 jcm-11-04831-f002:**
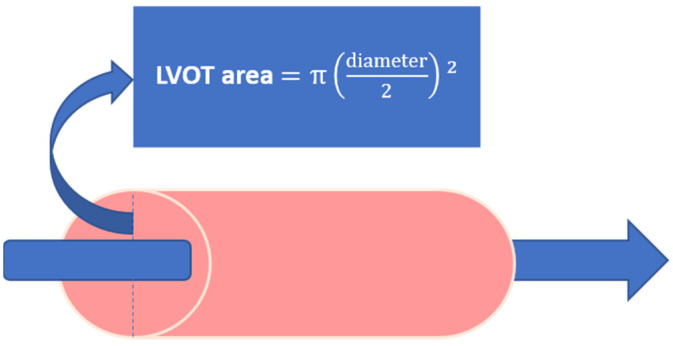
LVOT area calculation based on the diameter of the LVOT. Abbreviations: LVOT, left ventricular outflow tract.

**Figure 3 jcm-11-04831-f003:**
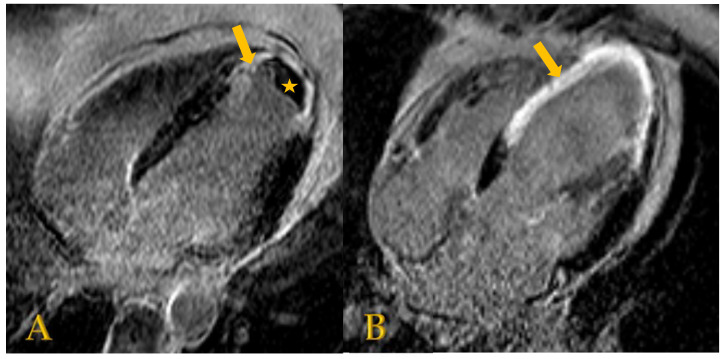
(**A**) CMR image showing a thrombus (asterisk) located in the apical left ventricular aneurysm in a patient with status after transmural infarction. Subendocardial late enhancement (arrow) is seen in the basal wall, lateral wall, and mid-portion of the septum. (**B**) CMR image showing transmural delayed enhancement extending from the basal to apical anteroseptal wall (arrow) in a patient with status after large transmural infarction of the anterolateral wall.

**Figure 4 jcm-11-04831-f004:**
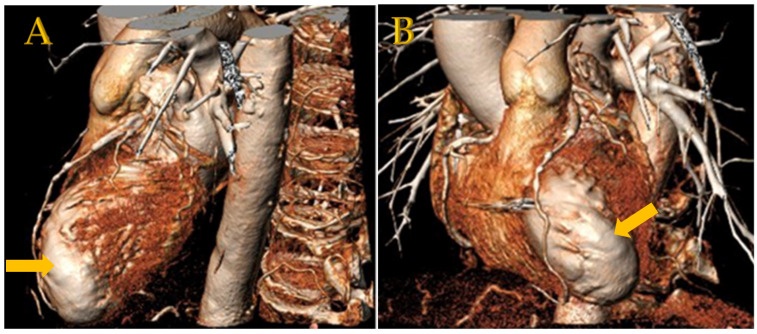
(**A**,**B**) Left anterior oblique (LAO) cranial coupes of CT-derived 3D reconstruction before LIVE procedure, made as part of procedural planning, revealing a prominent scarred aneurysm of the left ventricle (arrows).

**Table 1 jcm-11-04831-t001:** Primary in- and exclusion criteria of previous studies.

	Naar 2021 (*n* = 23)	Klein 2019 (*n* = 89)	Loforte 2019 (*n* = 7)	Klein 2019 (*n* = 9)	Wang 2021 (*n* = 26)
Inclusion criteria					
- LVEF	15–45%	15–45%	<35%	<40%	<40%
- NYHA class	II–IV	II–IV	III–IV	II–IV	II–IV
- OMT for ≥90 days	+	+	+	NR	+
- Transmural scarring	+	+	+	+	+
- A- or dyskinesia in the anteroseptal, anterolateral or apical region	+	+	+	+	+
- Left ventricular end-systolic volume index (LVESVI) ≥ 60 mL/m^2^	−	+	+	−	+
Exclusion criteria					
- Intracardiac thrombus	+	+	+	+	+
- Myocardial infarction within 90 days before the procedure	+	+	+	+	+
- Systolic pulmonary arterial pressure >60 mmHg on echocardiography or severe RV dysfunction	+	+	+	+	−
- Contraindication to open-heart surgery in case of conversion	+	−	+	+	−
- Cardiac valve disease that necessitates repair or replacement	−	+	+	+	−

Abbreviations: LVEF, left ventricular ejection fraction; LVESVI, left ventricular end-systolic volume index; mmHg, millimeter of mercury; NYHA, New York Heart Association; OMT, optimal medical therapy; RV, right ventricle.

**Table 2 jcm-11-04831-t002:** Study characteristics of the included studies.

Study ID	Naar 2021	Klein 2019	Loforte 2019	Klein 2019	Wang 2021
Country of Origin	Czech Republic	11 countries	Italy	The Netherlands	China
Study period	September 2013–March 2019	August 2010–March 2016	January 2015–November 2018	October 2016–July 2017	January 2017–January 2019
Study design	Prospective mono-center single-arm study	Prospective multi-center single-arm study	Mono-center single-arm study	Prospective multi-center single-arm study	Prospective mono-center single-arm study
Surgical technique	Hybrid *	Non-hybrid * Hybrid *	Hybrid *	Hybrid *	Hybrid *
No of patients	23	89	7	9	26
No of hybrid patients	23	35	7	9	26

* Hybrid: Hybrid transcatheter and minimally invasive surgical remodeling procedure on the beating heart utilizing the second generation Revivent TC system (BioVentrix, Inc., San Ramon, USA). * Non-hybrid: First-generation BioVentrix Revivent TC system on beating heart requiring median sternotomy.

**Table 3 jcm-11-04831-t003:** Baseline characteristics of previous studies.

Reference	Naar 2021	Klein 2019	Loforte 2019	Klein 2019	Wang 2021	Total
Patients (*n*)	23	35	7	9	26	100
Age (years)	59 ± 11	63 ± 10	72 ± 9	60 ± 8	58 ± 13	61 ± 11
Male (%)	65%	91%	71%	89%	-	81%
NYHA class (1–4)	2.3 ± 0.5	2.6 ± 0.5	3.4 ± 0.6	2.7 ± 0.4	2.7 ± 0.6	2.6 ± 0.5
6MWT (m)	381 ± 103	365 ± 90	-	-	369 ± 40	371 ± 82
LVEF (%)	32 ± 7	30 ± 8	23 ± 8	28 ± 8	36 ± 9	31 ± 8
LVESVI (mL/m^2^)	73 ± 27	75 ± 32	93 ± 11	53 ± 8	85 ± 26	76 ± 27
LVEDVI (mL/m^2^)	107 ± 27	110 ± 39	137 ± 20	75 ± 23	108 ± 33	108 ± 33
TR (0–4)	0.6 ± 0.6	-	-	0.5 ± 0.6	-	0.6 ± 0.6
MR (0–4)	1.2	1.1 ± 0.9	-	0.4 ± 1.2	-	1.0

Values are mean ± SD or *n* (%). Abbreviations: LVEDVI, left ventricular end-diastolic volume index; LVEF, left ventricular ejection fraction; LVESVI, left ventricular end-systolic volume index; MR, mitral regurgitation; NYHA, New York Heart Association; TR, tricuspid regurgitation; 6MWT, six-minute walk test.

**Table 4 jcm-11-04831-t004:** Procedural data.

Reference	Naar 2021	Klein 2019	Loforte 2019	Klein 2019	Wang 2021	Total
Patients (*n*)	23	35	7	9	26	100
Anchor pairs	2.9	-	3.0 ± 0.9	2.6 ± 0.7	2.7 ± 0.7	2.8
Operating time (min)	204 ± 50	-	195 ± 115	21 ± 100	304 ± 69	218 ± 74
Conversion to full sternotomy (*n*)	1 (4%)	0 (0%)	1 (14%)	1 (11%)	0 (0%)	3 (3%)
Re-operation (*n*)	2 (9%)	0 (0%)	0 (0%)	1 (11%)	0 (0%)	3 (3%)
Procedure-related mortality	2 (9%)	3 (9%)	0 (0%)	0 (0%)	1 (4%)	6 (6%)
ICU stay (days), median (IQR)	-	2 (2–5) *	4 (1.5–12.5) *	2 (1–46)	-	-
Hospital stay (days), median (IQR)	-	9 (7–15) *	14 (11.5–32) *	9 (3–57)	-	-
Ventilation time (hours), median (IQR)	-	9.5 (7–15.5) *	4 (4–17.5) *	-	-	-

Values are mean ± SD, median (IQR) or *n* (%). Abbreviations: ICU, intensive care unit; IQR, interquartile range; * calculated from (Appendix A) patient data. Wang: operation to general wards 6.9 ± 2.8 days.

## Data Availability

No new data were created in this study.

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
