# Peer review of "State-of-the-Art Review: Technical and Imaging Considerations in Hybrid Transcatheter and Minimally Invasive Left Ventricular Reconstruction for Ischemic Heart Failure"

_jcm, 2022, doi:10.3390/jcm11164831_

Round 1
Reviewer 1 Report
Overall, this review article is written for a comprehensive understanding of the individual procedural steps from both a cardiological and surgical point of view. The work can add to the field.
A number of minor concerns are outlined below.
1. Please add more wording on subheadings to help the reader easier understand and gain from each paragraph.
2. Please rewrite the subtitles in the paragraph “3. LIVE therapy stitching technique: what to stitch?” ( 3.1. Septal involvement: Hybrid RV-LV +/- External RV-LV (Antonius Stitch) +/- 108 External LV-LV +/- LV purse-string; 3.2. 3.2. External RV-LV +/- external LV-LV +/- LV purse-string; 3.3. External LV-LV + LV purse-string.) to help readers with a general biomedical background understand easily.
The cited references in several paragraphs are missing, such as in 3.1 from line 108 to line 1120; 3.3. from line 128 to line 132; 4.4 from line 218 to line 219; 5.1 from line233 to line 237; 5.8. from line 283-line 286; 5.9. and 5.10 from lines 288 to line 308; 5.12 from line318 to line 323; 6.1 from 328 to 340; 6.2 from 343 to line 353. 6.4 from 384 to line 398.
Author Response
1. Please add more wording on subheadings to help the reader easier understand and gain from each paragraph.
Answer: Based on your useful suggestion we have expanded the subheadings to make this manuscript easier to understand for all readers.
2. Please rewrite the subtitles in the paragraph “3. LIVE therapy stitching technique: what to stitch?” ( 3.1. Septal involvement: Hybrid RV-LV +/- External RV-LV (Antonius Stitch) +/- 108 External LV-LV +/- LV purse-string; 3.2. 3.2. External RV-LV +/- external LV-LV +/- LV purse-string; 3.3. External LV-LV + LV purse-string.) to help readers with a general biomedical background understand easily.
Answer: We agree with the reviewer that the titles in paragraph 3 are difficult to read for readers with a general biomedical background and have therefore changed all titles in paragraph 3.
3. The cited references in several paragraphs are missing, such as in 3.1 from line 108 to line 1120; 3.3. from line 128 to line 132; 4.4 from line 218 to line 219; 5.1 from line233 to line 237; 5.8. from line 283-line 286; 5.9. and 5.10 from lines 288 to line 308; 5.12 from line318 to line 323; 6.1 from 328 to 340; 6.2 from 343 to line 353. 6.4 from 384 to line 398.
Answer: We want to thank the reviewer for noticing the missing information. We have now added all missing references. Of note, in paragraph 3.1, 3.3, 5.1, 5.10 and 5.12 we have described our own technique as applied in our department in the St. Antonius Hospital. As this technique has not been described in previous literature, there are no references in these paragraphs.
Reviewer 2 Report
Romy R.M.J.J. Hegeman and colleagues summarized knowledge of the technical considerations and adequate use of multimodality imaging of the less invasive ventricular enhancement (LIVE) procedure for ischemic heart failure. In particular, the details of procedure process, pre- and post-operative imaging modalities were very easy to understand and helpful in clinical practice.
Overall, the findings are interesting, and the contents appear to be of good quality, but I provided few major and minor comments below:
Major comment
I feel it’s very informative that the comparison of echocardiographic and functional outcomes of each studies are summarized in table.
Minor comments
In Table1, ‘1’ or ‘0’ should be replaced with ‘+’ or ‘-’.
In page11 line 364, does Table2 need to be replaced with Table3? Table2 does not show baseline characteristics of previous studies.
In page12 line 383, does Table3 need to be replaced with Table4? Table3 does not show procedural data of each studies.
Author Response
Major comment
I feel it’s very informative that the comparison of echocardiographic and functional outcomes of each studies are summarized in table.
Answer: We want to thank the reviewer for this helpful remark. We have summarized all echocardiographic and functional data in two separate Tables. Because the tables are rather large, we have added a Supplementary file that includes these two Tables.
Minor comments
1. In Table 1, ‘1’ or ‘0’ should be replaced with ‘+’ or ‘-’.
Answer: We agree with the reviewer and have adjusted Table 1
2. In page11 line 364, does Table2 need to be replaced with Table3? Table2 does not show baseline characteristics of previous studies.
Answer: We would like to thank the reviewer for this comment, indeed Table 2 should be Table 3 in page 11 line 364. We have changed the titles accordingly.
3. In page12 line 383, does Table3 need to be replaced with Table4? Table3 does not show procedural data of each studies.
Answer: We agree that Table 3 should actually be Table 4 in page 12 line 383 and have changed it accordingly.
Round 2
Reviewer 1 Report
No more question
Reviewer 2 Report
Thank you for responding to our comments. The authors have substantially improved the manuscript following suggestions from the previous reviews. I'm satisfied with the author's sincere efforts and the revised manuscript.